# Effect of Amniotic Injection of N-Carbamylglutamate on Meat Quality of Broilers

**DOI:** 10.3390/ani10040576

**Published:** 2020-03-30

**Authors:** Feng-dong Zhang, Jing Wang, Hai-jun Zhang, Shu-geng Wu, Jing Lin, Guang-hai Qi

**Affiliations:** Risk Assessment Laboratory of Feed Derived Factors to Animal Product Quality Safety of Ministry of Agriculture & Rural Affairs, and National Engineering Research Center of Biological Feed, Feed Research Institute, Chinese Academy of Agricultural Sciences, Beijing 100081, China

**Keywords:** NCG, amino acid, pectoral muscle, meat quality, antioxidant

## Abstract

**Simple Summary:**

N-carbamylglutamate (NCG) is a key activator of endogenous arginine production, which plays a vital role in meat quality and antioxidant performance. To the best of our knowledge, there is no study about NCG enhancing the meat quality of broilers. The present research was aimed at exploring the effects of amniotic injection of NCG on meat quality of pectoral muscle in broilers. The data showed that the in ovo feeding (IOF) of NCG enhanced the arginine content, improved the nutritional properties, enhanced the antioxidant capacity, and improved the meat quality in the pectoral muscle of broilers. In summary, amniotic injection of NCG on day 17.5 of incubation could be an effective and novel approach to improving the meat quality of broilers.

**Abstract:**

The current study was performed to determine the influence of amniotic injection of N-carbamylglutamate (NCG) on meat quality of pectoral muscle in broilers. A total of 792 alive broiler embryos at 17 d of incubation were assigned to three treatments randomly (non-injected control, saline-injected control, or NCG-injected treatment). The two injection treatments were an injection with 0.1 mL 0.85% aseptic saline alone or containing 2 mg NCG per egg at 17.5 d of incubation. After hatching, 72 healthy male chicks were selected from each treatment and housed in six pens for a 42 day feeding study. Pectoral muscles from six 42-day-old broilers were collected from each treatment group and were dissected for meat quality assays. The results showed that arginine contents in pectoral muscle in either free or hydrolytic form in the NCG group were higher than those in the non-injection control group (*p* < 0.05). In comparison to the non-injection or saline-injection control groups, NCG injection resulted in a lower lactic acid content in pectoral muscle (*p* < 0.05). Muscular antioxidant capacity in the NCG group was higher, as evidenced by the higher activity of catalase and glutathione peroxidase and lower content of malondialdehyde (*p* < 0.05). In addition, the group of in ovo administration of NCG had decreased drip loss and increased crude fat content in pectoral muscle in comparison to those of either control group (*p* < 0.05) and had enhanced crude protein content compared to that of the saline-injection control group (*p* < 0.05). Briefly, these results indicate that amniotic administration of NCG in the late incubation phase increased the arginine content, improved the nutritional properties, enhanced the antioxidant capacity, and improved the meat quality in the pectoral muscle of broilers. Amniotic injection of NCG may serve as a novel approach to improving the meat quality of broilers.

## 1. Introduction

Recently, much attention has been paid to broiler meat due to its characteristics of low fat, high protein, and richness of functional peptides [1], which are all favored by health-conscious consumers. The nutritional supply is a vital element affecting the quality and composition of poultry meat [2]. As modern broiler chickens grow faster and the time-to-market becomes shorter, the embryonic development in the hatching phase becomes more and more important [3]. Therefore, the nutritional supply to broiler embryos may be a beneficial pathway to enhance the quality and composition of poultry meat.

In ovo feeding (IOF) is an emerging technology to administer exogenous nutrients into the hatching eggs for better embryonic development and postnatal growth exhibition. Preferably, the amnion of the late-term embryo (17.5–19 d of incubation) is the ideal site for nutrients administration to enhance growth performance and energy reserves [4,5,6]. In this context, in order to obtain favorable broiler meat quality, the modulation of the embryo nutritional supply in the incubation stage is a promising solution. Indeed, evidence has shown that administration of equol, L-ascorbic acid, or synbiotics in the incubation phase may effectively improve broiler meat quality [3,7,8,9]. Since the risk for protein catabolism in late embryos exists [10], it is interesting to probe the efficacy of IOF of amino acids on meat quality of market size broilers.

Arginine (Arg), one of the nutritionally essential amino acids in birds, had been used in the poultry diet to make meat quality better and increase antioxidant defense under normal circumstances [11]. N-carbamylglutamate (NCG), a stable activator for endogenous Arg synthesis, has been demonstrated to enhance Arg concentration, stimulate protein synthesis in skeletal muscle, [12] as well as improve meat quality in pigs [13] and sheep [14]. Up to now, the application of NCG in broilers is scarce. The only available study regarding NCG in broilers showed that diet supplementation of NCG in Chinese yellow-feathered broilers positively affected growth performance, tissue development, and blood parameters [15]. Whether the exposure of NCG in the incubation phase will exert positive effects on growth and meat quality in broilers is still unknown. Based on the above literature, exploring the influence of IOF of NCG on the meat quality of broilers could be valuable. In addition, it was reported that some nutrients or bioactive substances exert the same functions irrespective of delivery route, as evidenced by the reduction of susceptibility of poultry to pulmonary hypertension syndrome with supplemental Arg performed in ovo or in the feed [16], and the positive effects on meat quality of broilers affected by prebiotics either by an in ovo or in water route [17]. However, IOF uses smaller amounts of supplements than diet supplementation, and the comprehensive effects of IOF technology are superior to diet addition for broilers in some supplements [16,18]. Thus, it is reasonable to hypothesize that in ovo delivery of NCG can convey profitable influences on broiler meat quality. The objectives of this research were to investigate the influences of amniotic injection of NCG on meat quality of broiler cocks as well as to demonstrate whether exposure to NCG modulated the glycolytic and free Arg content, nutritional properties, and antioxidant capacity in the pectoral muscle.

## 2. Materials and Methods

### 2.1. Incubation

Fifty fertilized broiler eggs with the mean weight of 58–62 g from Ross 308 breeder were randomly assigned to every replicate. A total of 18 replicates were randomly placed in a microcomputer automatic incubator (37.8 ± 0.1 °C, 60%). After injection (at 17.5 d of incubation), hatching eggs were diverted to the hatching basket, which was placed in a microcomputer automatic hatcher (37.2 ± 0.1 °C, 70%) until chicks were hatched.

### 2.2. Treatment Solutions and Injection Procedure

NCG of 2 g, which was accurately weighed, was dissolved in 100 mL aseptic saline (0.85%) to prepare the NCG solution. All solutions were passed through a syringe filter membrane for sterilization, and then were drawn into 1 mL syringes pending for injection.

After 17.5 d of incubation, the fertile eggs were candled and the sites of injection were marked. Subsequently, 0.1 mL saline or NCG solution were injected into the amniotic cavity of eggs by a 2.54 cm, 22-gauge needle possessing a short beveled tip from the large end of the egg. To make sure the needle thrust entirely into the amnion, the whole needle was inserted into the large end of the egg. The injection site remained open. Based on our previous study, the accuracy of amnion injection is more than 95%. The routine procedure of checking the success of amnion injection was done by coomassie brilliant blue dye injection and autopsy observation.

### 2.3. Bird Housing and Feeding

Upon hatching, 72 healthy male chicks of each treatment were randomly assigned to six replicates for a 42-day grow-out feeding. A corn–soybean meal diet for starter (1–21 d) or grower (22–42 d) was formulated based on the National Research Council (1994). The starter diet contained 22% crude protein, 2950 kcal/kg metabolizable energy (ME), 1.15% digestible lysine, 0.85% digestible methionine + cystine, and 0.75% digestible threonine. The grower diet contained 20% crude protein, 3050 kcal/kg ME, 1.05% digestible lysine, 0.79% digestible methionine + cystine, and 0.69% digestible threonine. Chicks had free access to feed and water. Chicks were raised in wire floor cages (cage size, 110 × 100 × 55 cm^3^) in a three-level battery under environmentally controlled room conditions in the Nankou CAAS experimental base (Beijing, China). All the animal management was in accordance with the Ross 308 broiler management guide (Aviagen 2015) [19]. After the experiment, the remaining broilers were quarantined and sent to a local slaughterhouse. All the experimental protocols (AEC-CAAS- 20190105) for this study were approved by the Animal Care and Use Committee of the Feed Research Institute of the Chinese Academy of Agricultural Sciences.

### 2.4. Sample Collection and Breast Weight Measurement

One healthy bird with the body weight (BW) similar to the mean value on day 42 of cage life was selected and sacrificed by jugular cutting for blood and muscle sample collection. After being plucked and eviscerated, the pectoral muscles were weighed. The left pectoral, which was snapped frozen using liquid nitrogen, was kept at −80 °C pending measurement of various physical parameters.

### 2.5. Hatchability and Growth Performance Measurement

On the day of hatching, the number of chicks in each replicate were counted to calculate the hatchability. Hatchability equals the ratio between the number of alive embryos at E 17 (17 d of incubation) and the number of chicks.

At the end of experiment, all broilers in each replicate were weighed and feed intake was recorded. The growth performance was expressed by average daily gain (ADG), average daily feed intake (ADFI), and F/G (ADFI/ADG).

### 2.6. Physical Parameters of Pectoral Muscle

The method of determining pH values was improved based on the foundation of Qi et al. (2018) [20]. Three different locations (cranial, medial, and caudal) within the medial surface (bone side) of the breast muscle were used for pH readings by a pH meter (CyberScan pH 310, Eutech Instruments Pte. Ltd., Singapore) at 45 min and 24 h postmortem. The ΔpH was calculated as pH 24 h − pH 45 min.

The method of determining the meat color was improved based on the foundation of Zhang et al. (2009) [21]. Meat color was measured three times using a Chroma Meter (Chroma Meter WSC-S, Shanghai Precision and Scientific Instrument Co., Shanghai, China). The parameters (L*, a*, b*) of color were shown in the CIELab system.

The method of determining dripping loss was improved based on the foundation of Zhang et al. (2009) [21]. Approximately 30 g of pectoralis major muscle (W1) was suspended in a sealed plastic case with precooling at 4 °C. After 24 h in a 4 °C refrigerator, filter paper was used to absorb the surface moisture of samples and they were weighed again (W2). Drip loss (%) = (W1− W2)/W1 × 100%.

The method of determining cooking loss was improved based on the foundation of Qi et al. (2018) [20]. Meat samples in zip-sealed polyethylene bags were boiled in a water bath to 80 °C (i.e., until the sample’s center was 80 °C). Then, the samples were brought down to ambient temperature by running them under water. The samples were then dried using paper towel and weighed again (W3). Cooking loss (%) = (W2 − W3)/W2 × 100%.

### 2.7. Glycolytic Metabolite Measurements

Measurements of muscle glycolytic potential (GP) used the method recorded by Zhang et al. [21]. Specifically, the sample (frozen muscle, 0.5 g) was placed in a 4.5 mL perchloric acid solution and homogenized for 3 min. To remove protein and other impurities, the homogenate was centrifuged (4 °C, 3000× *g*, 10 min). To measure lactate in the supernatant, we used spectrophotometry, according to the instructions of the lactic acid determination kit (Nanjing Jiancheng Bioengineering Institute). In acetate buffer (pH 4.8, 55 °C, 2 h), glycogen was enzymatically hydrolyzed to glucose by amyloglucosidase (A7420, Sigma-Aldrich Inc., St. Louis, MO). Then, for measurement of glucose, we used a commercial glucose oxidase kit (Shanghai Rong Sheng Biotech Co. Ltd., Shanghai, China). GP = 2 × [glycogen] + [lactate].

### 2.8. Determination of the Antioxidant Index in Muscle

T sample (frozen muscle, 0.5 g) was put in 4.5 mL of a 0.8% saline solution and homogenized for 3 min. After centrifugation (4 °C, 3000× *g*, 10 min), supernatant fluid was collected for measurement of the activities of total superoxide dismutase (T-SOD), catalase (CAT), and glutathione peroxidase (GSH-Px) and of the volume of malondialdehyde (MDA). All the indicators were determined following by the manual provided with the kit (Nanjing Jiancheng Bioengineering Institute).

### 2.9. Hydrolytic Amino Acids Analysis

The method of determining hydrolyzed amino acids was improved based on the foundation of Nasr et al. [22] Briefly, each fat-free sample (0.5 g) was placed in a nitrogen-filled sealed tube and was hydrolyzed with 10 mL HCl (6 M) in an oven (110 °C, 24 h). The hydrolysate was adjusted to a volume of 500 mL with chromatographic water before passing through a 0.2 μm filter membrane. The concentrations of amino acids labeled with phthalaldehyde were determined at 338 nanometer derivative wavelength using high performance liquid chromatography (HPLC) with Shimadzu SPD-20A. Mobile phase A is an organic phase (acetonitrile/methanol/chromatographic water = 4.5:4.5:1) and mobile phase B is an inorganic phase (4.5 g disodium hydrogen phosphate dodecahydrate, 4.75 g sodium tetraborate decahydrate, 1000 mL chromatographic water). The flow rates of the mobile phases are shown in Table 1.

### 2.10. Free Amino Acid Analysis

The method of determining free amino acids was improved based on the foundation of Hartman et al. [23] Briefly, the sample (frozen muscle, 0.5 g) was placed into a saline solution (ice-cold, 4.5 mL, 0.8%) to homogenize. After centrifugation (4 °C, 9000× *g*, 10 min), 300 μL supernatant was mixed with 300 μL 1.5 M perchloric acid, left to stand for 10 min, then mixed with 150 μL of a 2 M potassium carbonate solution and left to stand for 20 min. The mixture was centrifuged and the supernatant filtered through a 0.2 μm membrane. Free amino acids were then determined by HPLC.

### 2.11. Chemical Composition of Pectoral Muscle

Muscle samples were put in a Petri dish and frozen and dried in a thermo freeze dryer (100 mbar pressure, −50 °C, Pilot7–12L, BIOCOOL) for 3 days. The Soxhlet extraction method was used to measure crude fat. Crude protein was measured using a Foss Kjeltec TM 8400 nitrogen/protein analyzer.

### 2.12. Statistical Analysis

All data analyses were implemented using SAS (version 9.4; SAS Institute, 2013). Means were separated for significance by Duncan’s multiple range tests. The data for meat quality, amino acids, antioxidants, glycolytic potential, and chemical parameters were analyzed using one-way ANOVA in SAS. Phenotypic correlations were performed using PROC CORR in SAS. The discrepancy was deemed statistically significant at *p* ≤ 0.05, and results are shown as means and SEM.

## 3. Results

### 3.1. Hatchability, Growth Performance, and Breast Muscle Weight

The effects of IOF of NCG on hatchability, growth performance, and breast muscle weight in broilers are presented in Table 2. There was no significant differences in all indices of hatchability, breast muscle weight, and growth performance among the three treatment groups (*p* > 0.05).

### 3.2. Hydrolytic Amino Acid Profile in Pectoral Muscle

Table 3 shows the hydrolytic amino acid contents of pectoral muscle in broilers. The hydrolytic Arg content increased significantly (*p* < 0.05) in the IOF of NCG group compared to that in the saline-injection and control groups. The total and branched-chain amino acid content in the NCG treatment group increased markedly compared with that of the saline-injection group (*p* < 0.05) but was not statistically different (*p* > 0.05) from that of the control group. The concentration of hydrolytic alanine, tyrosine, isoleucine, and delicious amino acids in pectoral muscle tended to increase, which was due to the IOF of NCG (0.05 < *p* < 0.1).

### 3.3. Glycolytic Potential

The effects of IOF of NCG on the glycolytic potential of breast muscle in broiler were presented in Table 4. Lower concentration of lactate was detected in NCG in ovo injection group than the other two groups (*p* < 0.05), but no statistical differences on Glycogen and GP existed (*p* > 0.05).

### 3.4. Muscular Antioxidant Parameters

The data of IOF of NCG on activities of antioxidant enzymes and MDA content in the pectoralis major muscle are shown in Figure 1. In pectoral muscle, the activity of CAT was markedly higher in the IOF of NCG group (*p* < 0.05). In comparison to saline-injected and control groups, the activity of GSH-PX was markedly enhanced and the content of MDA was significantly decreased in the NCG group (*p* < 0.05).

### 3.5. Meat Quality Parameters

Table 5 shows the influences of IOF of NCG on meat quality. In pectoral muscle, the meat color, pH, and cooking loss did not significantly differ among all treatments (*p* > 0.05). Drip loss in the NCG-injection group decreased markedly compared to that in the saline-injected or control groups (*p* < 0.05).

### 3.6. Chemical Parameter of Pectoral Muscle

The influence of IOF of NCG on chemical constituents of the pectoralis major muscle in broilers are shown in Table 6. Compared with the non-injected or saline-injected control groups, IOF of NCG caused markedly lower moisture and higher crude fat content in pectoral muscle (*p* < 0.05). Furthermore, the NCG treatment group significantly increased the crude protein content and the protein/moisture ratio in the pectoralis major muscle compared to that of the saline-injection group (*p* < 0.05).

### 3.7. Correlations among Glycolytic Potential, Antioxidant Ability, and Meat Quality Parameters

The correlation coefficients among parameters of chicken pectoralis major muscle are shown in Table 7. The MDA content was significantly and negatively correlated with fat content (r2 = −0.591; *p* < 0.05) and closely and positively correlated with drip loss (r2 = 0.626; *p* < 0.05). Crude protein content correlated positively with GSH-PX activity (r2 = 0.520; *p* < 0.05) and negatively with moisture content (r2 = −0.793; *p* < 0.01).

### 3.8. Free Amino Acids Profile in Pectoral Muscle

The data of IOF of NCG on free amino acids of pectoralis major muscle in broilers are shown in Table 8. Compared with the non-injection control group, IOF of NCG markedly enhanced the content of free Arg in pectoral muscle (*p* < 0.05). Among all treatments, IOF of NCG significantly enhanced the content of free Ala and DAA (*p* < 0.05).

## 4. Discussion

N-carbamylglutamate (NCG) is a metabolic activator that stimulates the endogenous synthesis of Arg. NCG participates in activation of pyrrol 5 carboxylate synthase (P5CS) and carbamyl phosphate synthase 1 (CPS-I), then promotes the synthesis of citrulline, thus stimulating endogenous synthesis of Arg by argininosuccinate synthase (ASS) and argininosuccinate lyase (ASL) [24]. The current study demonstrated that early exposure of NCG in the hatching phase may increase the Arg content in muscle of market size broilers. The increase of Arg had been revealed in the muscle of finishing pigs fed NCG-containing diets [13]. Because Arg is involved in the metabolism of glucose and amino acids [25], protein synthesis [26], and antioxidant capacity [27], it is reasonable to predict that the increased Arg induced by NCG exposure may affect glycolysis and antioxidant ability in broiler chickens.

It is through the process of energy depletion that muscle turns to meat. In early postmortem, ATP is supplied by the decomposition of phosphocreatine (PCr) and then by the decomposition of glycogen to lactate after the 70% of the PCr is used up [28]. Glycolysis in muscle is a vital action in the postmortem period, when the oxygen supply for oxidative metabolism is lost due to exsanguination [29]. The end product of glycolysis is lactic acid, whose content could serve as an indicator to reflect the process of glycolysis. In the current study, the reduced lactic acid content may be due to the delayed occurrence of glycolysis in the presence of the stimulated production of creatine from the increased Arg, which has been observed in early studies [30,31]. Lower lactic acid would contribute to the improvement of meat quality.

Meat quality could be greatly affected by the postmortem redox state [32]. Metabolic processes in body and muscular tissue cause the formation of different reactive oxygen species (ROS), peroxides, singlet oxygen, and free radicals [33,34]. Excessive amounts of ROS or free radicals in muscles cause great damage to cell walls, functional proteins, and lipids, thus leading to the reduction of meat quality [35]. In organisms, the antioxidative defense system, including enzymatic and non-enzymatic antioxidants, can prevent the production of ROS [36]. Enzymatic antioxidant systems of superoxide dismutases (SOD), glutathione peroxidases (GSH-PX), and catalase (CAT), effectively decrease the accumulation and production of oxides [37], further reducing the damage to cells and tissues. The dismutation of superoxide radicals (O_2_−) is promoted by SOD to form oxygen and hydrogen peroxide (H_2_O_2_), then, H_2_O_2_ is reduced to glutathione and water by GSH-PX, and to oxygen and water by CAT [38]. In our study, the increasing activity of CAT and GSH-PX indicated that IOF of NCG would benefit the enzymatic antioxidant system. Malondialdehyde (MDA), a biomarker for the evaluation of lipid peroxidation, usually increases in muscle with elevated oxidative production [39]. The relatively lower concentration of MDA means that the muscle cells were more intact and the oxidation rate of lipids was slower in pectoral muscle of NCG-treated birds.

A major finding in our study was that injection of NCG in amnion markedly enhanced the protein content in pectoral muscle. The elevation of protein synthesis was also revealed in a previous study in piglets, in which NCG supplementation in diet improved the absolute rates of protein synthesis in skeletal muscle by increasing the level of Arg [12]. Enhancement of protein synthesis in skeletal muscles has been shown to be stimulated by Arg through the mTOR signaling pathway [40]. In addition, Arg supplementation increased fat content through the increase of PPARγ expression and stimulation of differentiation and proliferation in bovine adipocytes [41]. In the current study, crude fat in pectoral muscle increased in the NCG in ovo supplementation group. The increase of protein and fat in pectoral muscle may be partly associated with the improvement of oxidation resistance. Previous research noted that free radicals of oxidized products could induce the increase of protein denaturation [42] and lipid superoxidation in muscle [43]. Our results showed that the increased crude fat content was closely correlated with the MDA content and protein content was closely related to GSH-PX, in pectoral muscle.

Water holding capacity (WHC) is a vital indicator of meat characteristics that affect processing and sensory qualities, and lower WHC is always linked with economic losses [44]. Dripping loss is an easy and important parameter to reflect the WHC. Lactic acid and reactive oxygen generated from the biochemical reactions in the course of muscle turning to meat may decrease the completeness of muscle cell membranes and break down the functional proteins in muscle [36], which would result in the increase of drip loss of muscle [45]. In the current study, the reduction of drip loss was coincident with the decrease of MDA content in the pectoral muscle. Similar to our results, simultaneous decline of drip loss and MDA content was observed in porcine longissimus dorsi [46]. These observations suggested that the reduction of oxidation products contributes to the reduction of drip loss in muscle. A previous study reported that decreases of glycolytic rate and lactic acid content may lead to enhancement of pH and decreased drip loss in pork [47]. However, our results did not show the tie between lactic acid content, pH value, and drip loss. The discrepancy was probably due to the large variations during the course of pH decline in practical situations [28]. It was suggested that pH values were closely connected with glycogen rather than the lactate concentration [48]. In the current study, the decreased drip loss may be mainly related to the enhancement of antioxidative status. The antioxidant vitamin E has been shown to maintain muscle cell membranes [49]. However, more investigation is needed to support this idea.

Meat flavor is also a vital index for meat quality for consumers, which is expressed by odor and taste. Free amino acids in muscle are important flavor-presenting substances and flavor precursors of chicken meat. The main substance of umami is Glu, and Ala, Ser, Thr, and Lys are the main substances of sweetness [50]. Among the free amino acids, Arg and Ala content account for a higher proportion and are key flavor amino acids. The Maillard process, which occurs between amino acids and reducing sugars, is the major course for producing meat-fragrant compounds [51]. The current research showed that IOF of NCG enhanced the content of free Arg and Ala, and the total delicious amino acids in pectoral muscle of broilers. The reason for this result may be that the NCG stimulated synthesis of Arg [24] and then the Arg increased content of Ala [52,53]. The results showed that amniotic injection of NCG could positively affect the flavor of pectoral muscle in broilers.

## 5. Conclusions

In conclusion, amniotic injection of NCG at 17.5 d of incubation in broiler fertile eggs aids in meat quality improvement in broilers, as evidenced by increased arginine (hydrolytic and free) content, improvement of antioxidant capacity, and decreased lactic acid content in breast muscle. In ovo injection may serve as a promising approach for meat quality modulation in broiler chickens. The mechanism by which NCG in ovo supplementation enhances arginine synthesis and muscle accumulation in broilers warrants further study, to benefit not only broiler producers but also the meat industry.

## Figures and Tables

**Figure 1 animals-10-00576-f001:**
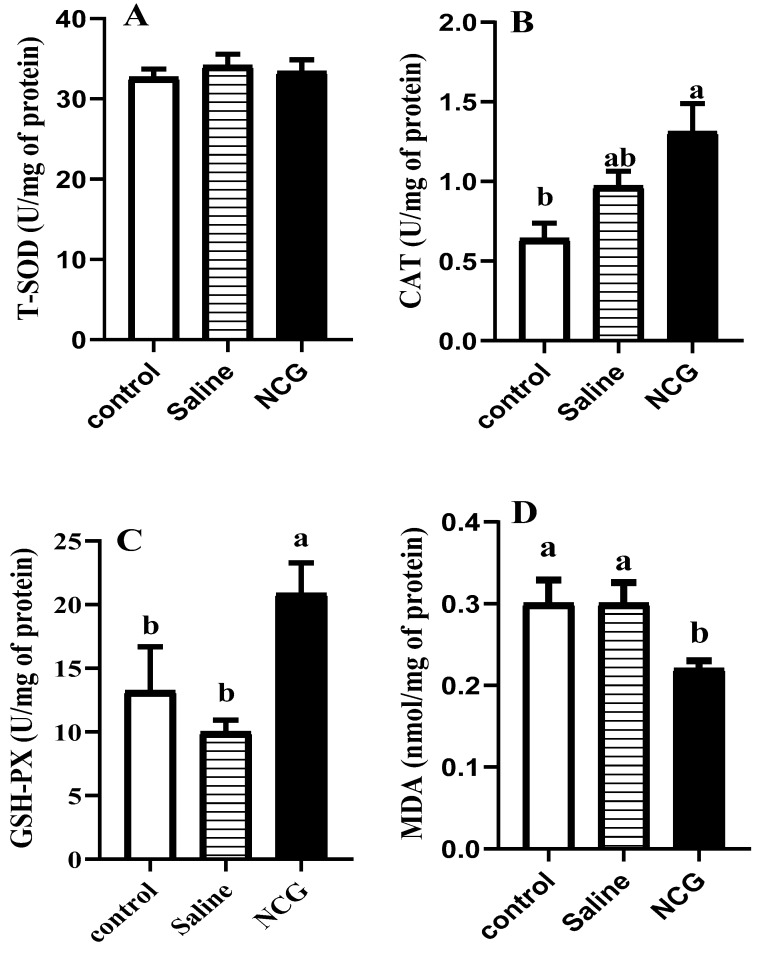
(**A**–**D**). Antioxidant activity of pectoral muscle in broilers. Values represent the mean + SEM (n = 6). Different superscript (a, b) indicate a significant difference (*p* < 0.05). Control = non-injected control group; Saline = saline-injected control group (8.5 g NaCl dissolved in 1 L distilled water); NCG = 2 mg NCG solution-injected group (20 g NCG dissolved in 1 L saline).

**Table 1 animals-10-00576-t001:** Mobile phase-gradient elution procedure.

Time (minute)	Mobile Phase A (%)	Mobile Phase B (%)	Flow Rate (mL/min)
0	95	5	1.6
6	90	10	1.6
8	90	10	1.6
10	84	16	1.3
23	60	40	1.0
30	50	50	1.6
31	0	100	1.6
34	0	100	1.6
35	95	5	1.6
38	95	5	1.6

**Table 2 animals-10-00576-t002:** Hatchability, growth performance, and breast muscle weight of broilers.

Items	Control	Saline	NCG	SEM	*p* Value
Hatchability (%)	88.19	90.63	91.67	0.872	0.261
42 d					
Breast weight (g)	423.50	425.33	413.00	9.479	0.863
BW (g)	2326.42	2307.16	2326.56	33.992	0.969
1–42 d					
ADG (g)	51.15	51.79	50.64	0.849	0.873
ADFI (g)	93.26	91.88	90.02	1.235	0.589
F/G	1.827	1.775	1.780	0.013	0.206

BW, body weight; ADG, average daily gain; ADFI, average daily feed intake; F/G, ADFI/ADG.

**Table 3 animals-10-00576-t003:** Hydrolytic amino acids profile of pectoral muscle in broilers.

Amino Acid (mg/g meat)	Control	Saline	NCG	SEM	*p* Value
Asp	38.42	36.55	38.03	0.871	0.680
Glu	52.74	50.86	52.66	0.548	0.301
Ser	14.16	13.64	14.24	0.156	0.251
His	13.81	13.49	13.28	0.212	0.608
Gly	14.77	14.04	14.53	0.145	0.108
Thr	15.09	14.64	15.46	0.216	0.324
Arg	40.53 ^b^	40.72 ^b^	43.56 ^a^	0.574	0.043
Ala	19.31	19.20	19.97	0.146	0.056
Tyr	11.37	11.29	11.96	0.141	0.098
Cys	0.46	0.47	0.42	0.051	0.930
Val	16.06	15.64	16.91	0.252	0.108
Met	9.42	9.03	9.22	0.135	0.519
Ile	16.25	15.80	17.24	0.267	0.068
Phe	14.48	14.40	14.56	0.236	0.967
Lys	30.16	29.56	30.81	0.253	0.130
Leu	27.32	26.48	27.86	0.350	0.284
TAA	334.36 ^ab^	325.81 ^b^	340.73 ^a^	2.447	0.033
EAA	183.80	178.26	184.39	1.911	0.374
DAA	180.27	175.18	182.99	1.507	0.093
LNAA	93.09	90.78	91.84	1.460	0.830
BCAA	59.63 ^ab^	57.92 ^b^	62.01 ^a^	0.650	0.024

^a,b^ Means in the same row with different letters are significantly different at *p* < 0.05. SEM, standard error of the means. TAA sum of total amino acids. DAA sum of the delicious amino acids, including Asp, Ser, Glu, Gly, Ala, and Arg. EAA sum of the essential amino acids, including Lys, Met, Thr, Ile, Leu, His, Arg, Val, and Phe. LNAA sum of the large neutral amino acids: Val, Ile, Leu, Phe, Tyr, and Met. BCAA sum of the branched-chain amino acids: Val, Ile, and Leu. Control = non-injected control group; Saline = saline-injected control group (8.5 g NaCl dissolved in 1 L distilled water); NCG = 2 mg NCG solution-injected group (20 g NCG dissolved in 1 L saline).

**Table 4 animals-10-00576-t004:** Glycolytic potential of pectoral muscle in broilers.

Items	Control	Saline	NCG	SEM	*p*-Value
Lactate (μmol/g)	96.28 ^a^	98.18 ^a^	84.56 ^b^	2.433	0.045
Glycogen ^1^ (μmol/g)	18.26	22.99	24.43	2.379	0.566
GP ^2^ (μmol/g)	132.80	144.16	133.42	5.259	0.628

^a,b^ Means in the same row with different letters are significantly different at *p* < 0.05. SEM, standard error of the means. ^1^ Includes glucose, glucose-6-phosphate, and glycogen. ^2^ Glycolytic potential = 2 × [glycogen] + [lactate]. Control = non-injected control group; Saline = saline-injected control group (8.5 g NaCl dissolved in 1 L distilled water); NCG = 2 mg NCG solution-injected group (20 g NCG dissolved in 1 L saline).

**Table 5 animals-10-00576-t005:** Meat quality of pectoral muscle in broilers.

Items	Control	Saline	NCG	SEM	*p* Value
45 min					
L*	42.98	42.84	41.61	0.731	0.726
a*	7.96	6.83	7.44	0.400	0.538
b*	10.50	12.01	10.28	0.502	0.328
pH	6.30	6.25	6.40	0.045	0.414
24 h					
L*	47.91	49.80	49.17	0.503	0.313
a*	5.67	5.23	4.73	0.234	0.278
b*	10.04	10.80	10.23	0.256	0.477
pH	5.94	5.85	5.92	0.022	0.212
ΔPH	0.359	0.401	0.477	0.050	0.654
Drip loss (%)	6.18 ^a^	5.05 ^a^	3.71 ^b^	0.341	0.004
Cooking loss (%)	26.23	25.19	23.86	1.401	0.696

^a,b^ Means in the same row with different letters are significantly different at *p* < 0.05. SEM, standard error of the means. Control = non-injected control group; Saline = saline-injected control group (8.5 g NaCl dissolved in 1 L distilled water); NCG = 2 mg NCG solution-injected group (20 g NCG dissolved in 1 L saline).

**Table 6 animals-10-00576-t006:** Chemistry composition of pectoral muscle in broilers.

Items	Control	Saline	NCG	SEM	*p* Value
Moisture (%)	68.37 ^a^	68.65 ^a^	64.43 ^b^	0.819	0.046
Crude fat (%)	2.05 ^b^	2.48 ^b^	3.65 ^a^	0.254	0.017
Crude protein (%)	25.84 ^ab^	23.02 ^b^	27.63 ^a^	0.802	0.040
Moisture/protein ratio	2.68 ^ab^	3.01 ^a^	2.37 ^b^	0.111	0.042

^a,b^ Means in the same row with different letters are significantly different at *p* < 0.05. SEM, standard error of the means. Control = non-injected control group; Saline = saline-injected control group (8.5 g NaCl dissolved in 1 L distilled water); NCG = 2 mg NCG solution-injected group (20 g NCG dissolved in 1 L saline).

**Table 7 animals-10-00576-t007:** Correlations among of glycolytic potential, antioxidant ability, and meat quality of breast meat.

Trait	Correlation Coefficient (r)							
	Lactate (μmol/g)	CAT (U/mg of Protein)	GSH-PX (U/mg of Protein)	MDA (nmol/mg of Protein)	Moisture (%)	Crude Fat (%)	Crude Protein (%)	Drip Loss (%)
Lactate (μmol/g)	1	−0.495	−0.443	0.237	0.460	−0.411	−0.350	0.184
CAT (U/mg of protein)		1	0.274	−0.448	−0.188	0.360	0.011	−0.484
GSH-PX (U/mg of protein)			1	−0.063	−0.432	0.223	0.520 *	−0.100
MDA (nmol/mg of protein)				1	0.181	−0.591 *	−0.126	0.626*
Moisture (%)					1	−0.439	−0.793 **	−0.064
Crude fat (%)						1	0.163	−0.419
Crude protein (%)							1	0.120
Drip loss (%)								1

* *p* < 0.05; ** *p* < 0.001.

**Table 8 animals-10-00576-t008:** Free amino acids of pectoral muscle in broilers.

Items	Control	Saline	NCG	SEM	*p* Value
Asp (mg/100 g)	18.55	17.50	17.59	0.431	0.571
Glu (mg/100 g)	5.76	5.52	5.36	0.151	0.588
Ser (mg/100 g)	3.38	4.17	3.78	0.249	0.456
His (mg/100 g)	1.46	1.50	1.25	0.117	0.670
Gly (mg/100 g)	6.86	8.23	6.16	0.540	0.299
Thr (mg/100 g)	3.50	3.94	3.75	0.251	0.793
Arg (mg/100 g)	19.57 ^b^	20.51 ^ab^	22.01 ^a^	0.424	0.049
Ala (mg/100 g)	70.37 ^b^	74.37 ^b^	95.60 ^a^	4.074	0.014
Tyr (mg/100 g)	1.94	2.31	1.64	0.147	0.178
Cys (mg/100 g)	0.68	0.84	0.57	0.059	0.180
Val (mg/100 g)	2.12	2.62	1.59	0.216	0.153
Met (mg/100 g)	1.13	1.29	0.90	0.084	0.155
Ile (mg/100 g)	1.06	1.37	0.75	0.135	0.174
Phe (mg/100 g)	1.55	1.80	1.39	0.140	0.524
Lys (mg/100 g)	3.92	5.90	4.00	0.420	0.086
Leu (mg/100 g)	1.88	2.53	1.46	0.219	0.133
EAA (mg/100 g)	36.19	41.45	37.10	1.532	0.346
DAA (mg/100 g)	124.48 ^b^	130.30 ^b^	150.50 ^a^	4.142	0.016
LNAA (mg/100 g)	9.68	11.91	7.73	0.886	0.157
BCAA (mg/100 g)	5.06	6.51	3.80	0.565	0.147
TAA (mg/100 g)	150.18	159.05	167.95	5.282	0.414
EAA/TAA (%)	24.24	26.38	22.11	0.936	0.181
DAA/TAA (%)	83.73	82.57	89.64	1.516	0.122

^a,b^ Means in the same row with different letters are significantly different at *p* < 0.05. BW, body weight; ADG, average daily gain; ADFI, average daily feed intake; F/G, ADFI/ADG.

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
