# Peer review of "Effect of Amniotic Injection of N-Carbamylglutamate on Meat Quality of Broilers"

_animals, 2020, doi:10.3390/ani10040576_

Round 1

Reviewer 1 Report

The manuscript displayed addressed the role of NCG in nutritional property, antioxidant capacity and meat quality of pectoral muscle of broiler. The study was appropriately designed and provides new insights into improving meat quality of broiler. It is suggested to introduce certain corrections:

Line 29. “NCG in ovo administration…” should be “NCG group in ovo administration…”.

Line 42. “importance” should be “important”.

Line 58-62. It is suggested to add some references to compare the differences between ovo feeding (IOF) technology and diet supplementation.

Line 64-68. Please state the experimental animal of reference 17.

Line 92-95. Please indicate how to treat the animals after the experiments and add the ethical approvement of experimental animals.

Line 98. “Calculating” should be “Calculate”.

Line 96-110. It is better to add the references in the part of physics parameter of pectoral muscle.

In the table 2, “ASP” should be “Asp”.

In the table 2, it is better to delete “2mg” in the boxhead.

Line 180. “a-c” should be “a-b”.

Table 5. There is no need to record the meat color at 45min after slaughter.

There is no need to include subtitles in the part of discussion.

Reviewer 2 Report

This paper tested effect of amniotic injection of N-cabamylglutamate on meat quality of boilers.

It is interesting and found good information. However, I found several point to improve this paper before publishing.

  1. Please be concerns about its formatting.
  2. A space is required between Number and units (i.e., 2 mg NC/ 100 g etc.)
  3. Is there a reason if you tested breast meat only?
  4. If so, how you are so sure it represent meat quality of chicks? Thanks.

Reviewer 3 Report

In general, significant works have been done to accomplish this research trial. However, the language in the current manuscript needs to be improved. I had some hard times to understand some of the sentences. Furthermore, the author needs to describe some of the crucial information in methodology as well as some of the critical details that are missing from the current manuscript:
1: Hatchability?
2. growth performance?
3. chicken breast weight, if any?

Furthermore:
Line 75: the number of eggs here did not match what was in the abstract.
Line 77: when is the injection?
Line 81: why chose this dosage? More details on how to perform the injection and the rate of successful injections.
Line 87: feed formulation and the nutrients requirement. Raising environment (floor pen, cage, or battery?)
Line 97: How many times that PH measurement is taken for each sample?
Line 155: what was used for mean separation?

Round 2

Reviewer 1 Report

After the revision, the quality of the paper has been improved. It is suggested to be published.